# Fraternal Twins: The Enigmatic Role of the Immune System in Alphaherpesvirus Pathogenesis and Latency and Its Impacts on Vaccine Efficacy

**DOI:** 10.3390/v14050862

**Published:** 2022-04-21

**Authors:** Barry T. Rouse, D. Scott Schmid

**Affiliations:** 1College of Veterinary Medicine, University of Tennessee, Knoxville, TN 37996, USA; 2Independent Contract Consultant, Littleton, CO 80125, USA

**Keywords:** herpes simplex virus, varicella zoster virus, latency, immunity, anti-herpesvirus vaccines

## Abstract

Although the establishment, maintenance and reactivation from alphaherpesvirus latency is far from fully understood, some things are now manifestly clear: Alphaherpesvirus latency occurs in neurons of the peripheral nervous system and control of the process is multifactorial and complex. This includes components of the immune system, contributions from non-neuronal cells surrounding neurons in ganglia, specialized nucleic acids and modifications to the viral DNA to name some of the most important. Efficacious vaccines have been developed to control both acute varicella and zoster, the outcome of reactivation, but despite considerable effort vaccines for acute herpes simplex virus (HSV) infection or reactivated lesions have thus far failed to materialize despite considerable effort. Given the relevance of the immune system to establish and maintain HSV latency, a vaccine designed to tailor the HSV response to maximize the activity of components most critical for controlling reactivated infection might limit the severity of recurrences and hence reduce viral transmission. In this review, we discuss the current understanding of immunological factors that contribute to HSV and VZV latency, identify differences between varicella-zoster virus (VZV) and HSV that could explain why vaccines have been valuable at controlling VZV disease but not HSV, and finish by outlining possible strategies for developing effective HSV vaccines.

## 1. Introduction 

It is evident that primates have harbored their herpes virus pathogens for millions of years and, perhaps not surprisingly, neither party now typically suffers more than minor inconvenience from this interaction. As leading HSV authority Bernard Roizman observed long ago: “unlike love herpes is forever”. However as with love, even in its passionate phase, there can be ups and downs with the virus causing episodes of discord that are almost invariably resolved satisfactorily—at least in hosts with a fully functional immune system. The viral party trick that mostly accounts for the harmony of the long-term tolerated relationship is the adoption by the virus of an alternative lifestyle called latency. During latency, the virus is not multiplying and not causing overt tissue damage, nor is it being spread to other hosts. Indeed, with both HSV and VZV latency, which mainly occurs in the peripheral nervous system, the latently infected neurons express few if any virally encoded proteins and thus would be expected to evade the attention of the immune system. However, the latent HSV virus does express RNA transcripts that are not found when the virus is undergoing productive viral replication. These so-called latency-associated transcripts (LAT) are somehow involved in the maintenance and perhaps the establishment of latency. While LAT is critical to this phenomenon, latency is far more complicated, involving multiple molecular and immunological co-factors. The topic has enthralled and caused debate among herpes virologists for decades and the definitive story has yet to be written. However, there are many excellent reviews composed by the main participants in the latency saga and several should be read to obtain a balanced idea of what is likely to be occurring [1,2,3,4]. LAT involvement seems to be a critical component, but viral mutants unable to produce the LAT transcripts can still establish and maintain latency in animal models. Such mutants, however, do have difficulty re-entering the productive replication cycle when exposed to stimuli that normally cause reactivation and that terminates latency in individual nerve cells [1]. A complicating issue is that mice are not men and many studies have been and continue to be performed in mice. Moreover, even studies in animal models employing their homologous alphaherpesvirus may not faithfully reflect how HSV sets up and maintains latency in mankind [4].

VZV latency was initially believed to differ substantially from HSV latency due to observations that as many as nine viral proteins were readily demonstrated in latently infected neurons [5]. The discovery of varicella latency transcript (VLT) implied that VZV latency probably more closely resembles that of HSV [6,7]. Whether or not any viral proteins are produced during VZV latency remains to be formally demonstrated. In addition, it was determined with more sophisticated technology that VZV, previously believed to produce no LAT transcript, did in fact express high levels of VZV LAT (VLT) during latency [6]. This brings up a common fact in science. It is generally more difficult to be confident about negative data than it is for positive findings. 

Explaining how alphaherpesviruses set up, maintain and breakdown latency has commanded the attention of (and not infrequently frustrated) the brightest minds in the herpes virology field. The truth is that, while impressive progress has been made toward understanding latency, we are nowhere close to putting a final period on the stories for either HSV or VZV. 

In this opinion article, our mission is to emphasize the role that the immune system may be playing during the establishment, maintenance and breakdown of latency. Along the way, we plan to shed light on the reasons why vaccines against the closely related human viruses HSV and VZV have had remarkably different results. Against HSV viruses HSV-1 and HSV-2, in spite of enormous effort, we still lack a vaccine that is in any meaningful sense effective. In contrast, against its VZV cousin, with far less effort and a dearth of experimental animal models to evaluate approaches, we enjoy effective prophylactic vaccines as well as vaccines that successfully diminish the risk for developing the painful curse of shingles. These vaccines are somehow either preventing or containing reactivation of VZV from latency. 

## 2. Review

### 2.1. The Highlights of HSV Pathogenesis

The two species of herpes simplex viruses are prevalent human pathogens around the world. Estimates are that 3.7 billion persons are seropositive for HSV-1 and 500 million for HSV-2 [8]. Both agents infect surface tissues, with HSV-1 characteristically the oro-labial mucosa and sometimes the eye and HSV-2 the genital surface. However, this location preference is changing with HSV-1 now a common genital infection in many societies and HSV-2 infections are also found on the face [9]. HSV infections result from close contact with the usually abraded surface. The virus primarily infects epithelial cells and most often causes an inflammatory reaction that is dominated initially by components of the innate immune system. These innate responses usually succeed in controlling the infection and resolving lesions, including interferons, neutrophils, macrophages, dendritic cells and natural killer cells [10]. Indeed, HSV infections are more severe and can be lethal in patients with genetic deficiencies of innate immunity, especially those involving the production of or response to interferons [11]. 

The adaptive immune response is also induced following infection and likely plays a role in resolving lesions. This point is emphasized when HIV patients with suppressed immunity are infected with HSV. Such patients often suffer severe and more prolonged herpetic lesions [12]. Suppression of T cell mediated responses appears to be more consequential than suppressed antibody production, a finding supported by observations of primary immunodeficiencies in humans and some animal models [13]. However, viral-specific antibodies resulting from a previous infection likely play a major role in palliating reinfection, although it has been well documented in reports on genital herpes from the Wald group that reinfection does occur [14]. Additionally, the HSV virus does possess several immune evasive mechanisms [15] that impair but do not entirely suppress antibody and T cell effector activity against the virus.

No patient infected with HSV succeeds in extirpating the virus from their body. Thus, inevitably in the course of primary infection the virus gains entrance to sensory nerve endings and is transported retrograde by physiological axonal transport mechanisms to the nerve cell body. Much is now known, thanks especially to the Enquist and Cunningham groups [16,17], about the transport process and it seems that viral parts rather than intact virions are transported. Once within the axon, the virion is inaccessible by the immune system and there are no known practical approaches to interrupt axonal transport. 

The end of the viral journey has received much investigation since the outcome differs from one neuron to another for reasons still being explored. Some neurons support a productive cycle, which leads to the destruction of the neuron and many nearby satellite cells. Other neurons signal the virus in some incompletely understood fashion to adopt the latency lifestyle, which means expression of LAT transcripts, but usually no viral protein production [1,2]. The latent state is not permanent at least for individually infected neurons among the large number affected. However, just as we cannot explain how the lion in the African savannah selects a victim from countless wildebeest candidates, we do not understand why reactivation on a single occasion from latency occurs in only a tiny fraction of latently infected neurons. There are many reviews that exhaustively cover the controversial topic of latency and multiple articles should be read to obtain a balanced understanding of the latency saga [1,2,3,4]. However, for those interested mainly in immunity, an excellent review was recently published that describes and debates in an even-handed way the influence of intrinsic, innate and adaptive aspects of immunity on the establishment, maintenance and breakdown of ganglionic latency [18]. In this opinion article, we only strive to provide some take home consensus messages about the role of immunity during HSV latency with emphasis on observations that might explain why controlling HSV infections with vaccines has largely failed, which is not the case for the fellow alphaherpesvirus VZV.

For many years the paradigm was that during HSV latency no viral proteins were made by infected neurons. This idea made it difficult to consider any role for the adaptive immune system in controlling latency. However, times and technology have changed and now it seems that minimal protein expression can occur either continuously or more likely sporadically in some latently infected neurons [18]. Immunologists welcomed this idea since it means that the immune system could be participating in latency decision events and that conceivably these could be usefully redirected by the armory of immune manipulators now available. 

Some years back, some investigators began looking at the trigeminal ganglion (TG) in mice infected in the eye with HSV. They made the surprising observation that a pronounced inflammatory response was present in the TG. This became evident a few days after ocular infection and could remain for a prolonged period [19,20]. Initially, the reaction was mainly composed of innate cells, but subsequently T cells, particularly CD8 T cells, dominated the inflammatory reactions [18]. In C57/B6 mice, the majority of these CD8 T cells were viral antigen-specific and many were shown to surround neurons. Subsequently, reports from the Verjans group, who had access to human cadavaric material showed that inflammatory cells were common in human trigeminal ganglia and that in some cases [19], T cells cloned from ganglia showed reactivity to HSV specific peptides [10].

These observations raise questions whether the viral-specific T cell response in the TG, known now to involve CD4 and CD8 T cells, plays a role in latency decision events. Such T cell or even antibody responses are likely to occur too late following primary infections to influence the initial decision between establishing a latent or lytic outcome in neurons. However, the adaptive responses do serve to limit the extent of neural infection and in preventing its spread to the CNS [21]. Thus, SCID mice infected with HSV invariably die of encephalitis, although if infected with a low viral dose their TG neurons can be shown to support a latent infection [22]. Studies on the extent of neuronal latency in the TG in knockout mouse models with compromised aspects of adaptive immunity aimed at defining the critical mechanisms involved have been difficult to interpret since such animals usually develop compensating immune activities [23].

Adaptive immune activities might not have an impact on initial events after primary neuronal infection, but the responses are likely to act to limit virus spreading to satellite cells during the initial period and after reactivation, perhaps resulting in the expansion of latency in the TG. It is also conceivable that once latency is established, immune T cells responding to stimulation from transiently antigen expressing neurons or satellite cells, particularly DC [24], could conceivably signal reactivating neurons to stop progressing to productive replication. If and how this latter effect could occur is not known, but if it does happen it is likely to be mediated by cytokines such as interferon gamma acting to modulate molecular events in the neurons that sustain latency. This might imply that immunomodulators could be designed to maintain the latent state and avoid the misery of reactivated lesions. Hope springs eternal when it comes to dealing with herpes recurrences!

Another useful function of the inflammatory reaction, especially the T cell response in the TG, could be to prevent the derepressed virus from any reactivating neuron from anterograde passage in the axon to infect the brain. The outcome can be encephalitis, often a fatal consequence when it happens with HSV infections [25]. Fortunately, encephalitis is a rare outcome with HSV infections in mankind, but a frequent event in monkeys infected with HSV as is infection of humans with the macaque alphaherpesvirus (MHV-1) [26]. There is at least circumstantial evidence that the immune response in latently infected TG participates in constraining virus from entering the brain. In our own studies for example, we could show that if the metabolic function of T cells was compromised by treating animals with 2-deoxy-D-glucose (2DG) soon after they received ocular infection with HSV, many of the animals developed encephalitis and needed to be euthanized [27]. We could show that there were fewer activated cells in the TG of 2DG treated animals and pronounced reduction in the number of IFN-γ-producing cells. We surmised that the compromised T cells were unable to prevent the virus from spreading anterograde to the brain, but could provide no empirical evidence for this conjecture. However, using an in vitro system of viral reactivation, we could show that metabolic inhibition of T cell function with 2DG served to accelerate reactivation [27].

Although herpes may be forever, neuronal latency for HSV in the natural host decidedly is not. Latency breaks down periodically in response to a range of stimuli, but never in all latently infected neurons at the same time. Potential triggers for reactivation include emotional stress, hormonal changes, UV light exposure, among others. The consequences of reactivation also vary in different individuals and at different times. On most occasions, reactivation is unnoticed and subclinical, but at other times is overt, often troublesome, with lesions occurring usually at the primary site of infection. We suspect the variables that impact on lesion severity and frequency are multiple and include the efficacy of local immunity at the peripheral site and the load of the virus that the immune system has to deal with. We know from the elegant work of Zhu in the Corey group that waiting close to the nerve endings from which virus emerges during reactivation are antigen specific CD8 T cells that are members of the resident memory constituency [28]. These cells are believed to limit the extent of viral replication [29]. Additional components of immune defense, such as antigen-specific CD4 T cells and other protective antiviral participants such as NK cells, are also involved in affecting the outcome of reactivation, but these have received less study. The mystery is why are the defense participants more effective at certain times, but not at others. One likely reason could be that the numbers or state of activation and function of the immune participants responsible for control could vary. Conceivably, the variability could be explained, at least in part, by differences in host metabolism and nutrient supply as has been suggested by Zhu and colleagues [30].

We need to provide mechanistic explanations for the variable outcome since then it may become possible to manipulate the local sites to recruit and expand the relevant participants to the site and make them function more effectively. We are enthralled by the prospect that this might be achievable using the recently developed mRNA vaccine approach using vaccines composed of multiple mRNA species. These components could include molecules that recruit immune effectors as well as those that activate and expand the appropriate specificities of T cells, which Koelle and colleagues have demonstrated are involved in local lesion control [31]. If the explanation involves metabolic factors, these might be readjusted by nutritional changes or by local administration of drugs that change metabolic events [32]. Should this therapeutic vaccine approach achieve success against HSV recurrences, we shall be one step closer to achieving the accomplishments of our VZV colleagues who have devised successful vaccines that blunt the consequences of VZV reactivation [31].

### 2.2. Immunity and VZV Latency

It has long been understood that VZV establishes latency in cranial and dorsal root ganglia (DRG), and that reactivation from that state is characterized by the intra-ganglionic transit of the virus to the skin, producing a dermatomally restricted rash illness. The study of the establishment, maintenance and reactivation of VZV from latency has proven very difficult for a number of reasons. Firstly, VZV is an extremely fastidious virus and can only be effectively propagated in culture in a limited number of human cells and, less successfully in a few animal cells. A second and related issue is that there are no practical animal models. For example, the cotton rat model is of limited use, since it is not possible to reactivate VZV from infected neurons, and this likely reflects an abortive rather than a latent infection [33]. Similarly, while embryonic guinea pig cells are among the few non-primate cells that can be productively infected with VZV (passage in guinea pig cells was part of the process for attenuating the Oka vaccine strain), and neuronal tissue can be infected with VZV, it has not been possible to stimulate reactivation of the virus from these tissues [34]. Studies of the closely related simian varicella virus (SVV) in African green monkeys and macaques have provided many useful observations, but this model also has limitations [35]. One animal model that has provided many useful insights has been the SCID-hu mouse. Fetal DRG can be implanted under the kidney capsule and has been shown to retain most of the characteristics of normal DRG. Infection of these implanted DRG with VZV resulted in a transient viremia that resolved completely within weeks [36]. VZV-infected ganglia could be detected months later. Since these mice lack an acquired immune system, this indicated that establishment of VZV latency and shut-down of neuronal gene expression are not dependent on components of acquired immunity. SCID mice do have intact activities of the innate immune system, such as NK cells. Infiltration of macrophages and NK cells into VZV-infected ganglia suggest that components of the innate immune system may be involved in controlling VZV replication in neurons [37]. Introduction of human VZV-immune T cells resulted in similar DRG infiltration to that observed in post-mortem-excised DRG. Efforts to study ganglia removed quickly post-mortem have been complicated by rapid initiation of viral transcription induced by hypoxia [38]. Consequently, much of the characterization of VZV latency, including the role of infiltrating immune cells, must be inferred by observations made for HSV, not a desirable situation.

As noted earlier, older studies supported the notion that VZV latency was fundamentally different than that of HSV-1 and HSV-2 [39]. It has now been established that neurons latently infected with VZV do indeed liberally express a transcript (VLT) that is antisense to the ORF61 gene, an important transactivator and homolog to HSV ICP0 [7]. In addition to its apparent role in VZV latency, VLT is also expressed in some lytically infected cells [6,7]. A microRNA (VZVsncRNA13) antisense to the leader sequence of ORF61 promotes VZV growth and spread in VZV-infected epithelial cells, possibly by inhibiting VLT [40]. To a lesser extent, the ORF63 gene is also transcribed in latently infected neurons, but no other VZV transcripts and no VZV proteins are detectable. Early reports of multiple VZV proteins in latently infected cells were called into question when it was discovered that several ascites-derived anti-VZV reagents cross-react with the blood group A antigens [41]. Moreover, analysis of the complete VZV transcriptome in trigeminal ganglia obtained immediately post-mortem among VZV latently infected subjects produced no viral transcripts for up to 9 hours post-mortem (39). Thus, current observations imply that expression of LAT and shutdown of viral transcription are consistent among the human alphaherpesviruses. Again, although the transient partial gene expression observed in HSV-infected neurons has not been demonstrated for VZV, it can probably be predicted that a similar bi-/multi-trigger requirement for emergence from latency is also the modus operandi for VZV latency.

Symptomatic reactivation of VZV leads to a dermatomally distributed painful rash illness called herpes zoster. It has been well established that the single greatest risk factor for developing herpes zoster is advancing age, and this in turn correlates with an age-related decline in the immune system [42,43,44]. While it can be argued that this risk factor reflects the relatively lower capacity of the geriatric immune system to contain reactivation, the actual defect(s) that permits the emergence of varicella from the latent state has not been identified. 

On the other hand, multiple immune cells infiltrate latently infected ganglia and remain resident during latency [45,46,47]. Both innate immune cells (notably NK cells and macrophages) and T cells can be found in latently infected ganglia. Unlike HSV, there has, as yet, been no success in determining whether or not the infiltrating T cells (both CD4+ and CD8+) are antigen-specific for VZV. Innate immune cells, specifically NK cells and macrophages may play important roles in establishing and maintaining latency [46,47]. VZV infection of fetal dorsal root ganglia in the SCID-hu mouse model leads to a transient uptick in viral replication, followed by a reduction in the level of viral DNA and restricted viral transcription [48]. This observation implies that innate immunity plays a role in establishing VZV latency. Satellite glial cells (SGC) and microglia obtained from trigeminal ganglia of cadavers are induced by toll-like receptor ligands for TLR1-6 to produce IL-6 and TNF-α, both of which have regulatory effects on T cells [49]. Intraganglionic cytokines produced following SVV or VZV infection enhance T cell activity, including upregulation of both MHC I and II antigens, and secretion of IFN-α and IL-6 [45,46,47]. In a study of human whole fetal dorsal root ganglia, it was shown that bystander uninfected neurons, but not VZV infected neurons, produce the chemokine CXCL10, which is presumably the impetus for drawing immune cells into the infected ganglion [50]. In SVV-infected monkey ganglia, IFN-I-stimulated genes are upregulated in the infected neurons themselves, and type I interferon expression is upregulated in VZV-infected human DRG. These findings implicate neurons as participants in the control of viral replication. It has also been shown in the SVV model that reduction in viral transcription is accompanied by the infiltration of CD8+ T cells [46].

An increase in MHC-II expression on satellite glial cells also implies a critical role for MHC-II-restricted T cell responses. Depletion of CD4+ T cells or CD8+ T cells following SVV infection produced sustained lytic viral gene expression in 30–60% of infected ganglia, indicating that T cells infiltrating ganglia contribute to the control of SVV replication and potentially to the establishment of latency [51].

One promising avenue for studying latency and reactivation for VZV has been the development of in vitro assays using cultured neurons. Two general types of systems have been employed: (1) non-stem cell-derived neurons and (2) stem cell-derived neurons [52]. The first category involves using cell cultures of human neurons obtained from ganglia removed post-mortem. This approach has the fatal limitation that the majority of human ganglia already carry latent VZV infections. As such, it is not possible to use these cells to study the establishment of latent VZV infection. This shortcoming could conceivably be overcome by using ganglionic tissue from aborted fetuses, since such cells would be unlikely to carry a latent infection [5,7,53]. However, given that the MRC-5 cell line, established more than half a century ago from the lung tissue of an aborted fetus and still under attack today by anti-vaccine and religious groups—an approach using human neuronal cell cultures from this source is certain to be deluged with ethical objections. It has also not been possible to achieve reactivation from latency with the release of infectious progeny in fully differentiated neuronal cells. The model has, however, played a crucial role in identifying the VZV VLT transcript and in establishing the presence of ORF63 transcript in latently infected cells [7]. 

The use of stem cell-derived neurons in tissue culture has led to more promising results. This approach offers a number of advantages, e.g., the cells can be expanded and maintained in culture and, derived from the same volunteer, do not exhibit the genetic heterogeneity of other in vitro systems [52,53]. Moreover, sufficient knowledge about the factors required to drive the development of neurons has made it possible to impel in vitro differentiation to produce specific neuronal subtypes [54,55,56,57,58]. Human embryonic stem cell (hESC)-derived neurons have already been successfully used to both establish VZV latency and to induce reactivation with the production of infectious VZV [54,56]. Not all stem-cell-derived neurons are capable of supporting a latent VZV infection. Induced pluripotent human stem cells (iPSC) could be productively infected with VZV, but it was not possible to establish a persistent infection in these cells [55]. If stem cell-derived human neuronal cells can be combined with other cells in culture to better mimic the microenvironment of ganglia, it may provide a valuable tool for evaluating the role of immune cells in establishing, maintaining and reactivating the latent virus. 

The immune system appears to play important roles in controlling the latent state, but of course the overall process is much more complicated. The LATs produced by both HSV and VZV play an important role [7]. In addition, extensive modifications to the viral genome by histone deposition, base methylation and acetylation produce a state of repressed transcription that requires orchestrated reversal before full reactivation can occur [58,59]. MicroRNAs have now been identified in these viruses with evidence that one or more of them are involve in regulating latency [60]. This is necessarily a multi-factor process, which probably explains periodic limited transcription and translation in HSV-infected neurons; these episodes represent abortive attempts to emerge from latency.

### 2.3. The Quest for an Efficacious HSV Vaccine

For multiple decades, we have searched for an effective vaccine approach to prevent and control HSV infections in mankind. The rationale for developing an HSV vaccine is predicated on several factors. It is estimated that half a billion people are genitally infected with HSV, and it is the leading cause of genital ulcer disease [61]. Available evidence indicates that HSV elevates the risk of acquiring and transmitting HIV [62]. Finally, there are currently no effective strategies for substantially impacting the incidence of HSV infection. There are two main strategies for developing an HSV vaccine, each of which is dependent on the desired public health impact of the vaccine. The first strategy is for a prophylactic vaccine. A prophylactic vaccine would be administered prior to HSV infection, with the goal of inducing a broad immune response, ideally including a neutralizing antibody and both CD8+ and CD4+ activities. The goals of a prophylactic vaccine would be to reduce genital ulcer disease, to prevent or reduce the extent of latent infection, and/or to reduce shedding and transmission. The ultimate and likely unattainable goal would be to prevent HSV infection altogether. The second strategy is developing a therapeutic vaccine. A therapeutic vaccine would be administered to persons with a history of symptomatic genital ulcer disease or virus reactivating in the eye that ultimately causes blindness. The goals of a therapeutic vaccine are the same as for a prophylactic vaccine with the obvious exception of preventing HSV infection [63]. 

Vaccine development generally proceeds through a series of experimental trials, failure in any one of which might result in the abandonment of the vaccine candidate. Initially, candidate vaccines are evaluated in one or more animal model systems to (1) establish that the vaccine is not toxic and (2) that it induces an immune response. This phase of vaccine evaluation typically involves a rather cursory look at the immune response and more often than not meets expectations with flying colors. Vaccines that successfully complete their evaluation in animals then proceed through as many as four trial phases in human volunteers. 

Modeling studies have provided some useful insights into the requirements for a putative HSV vaccine [63]. For example, an HSV vaccine would not be required to simultaneously impact susceptibility to infection, disease severity and likelihood to shed and transmit the virus to be effective at the population level. Elaborating on that point, breakthrough effects of a prophylactic vaccine are most important when vaccine susceptibility effects are limited. In addition, catch-up vaccination is likely to be important to optimal vaccine impact. In populations with high prevalence of HSV-1, a vaccine needs to be effective in persons already HSV-1-infected. Last but not least, even if a vaccine is effective in only one sex, it could provide herd immunity for the other sex assuming that coverage levels are sufficiently high. A modeling study of HSV-2 and HIV transmission in sub-Saharan Africa concluded that a prophylactic vaccine just 75% effective against infection would be expected to reduce HSV-2 incidence >55% over 10 years, assuming 50% coverage of 14-year-olds with catch-up up to age 29. It was also predicted to prevent approximately 10% of HIV infections in the same time frame [64].

The track record for HSV vaccines has been perfectly abysmal. At least 20 vaccines have been attempted, with some still in various stages of trial. Without going into detail, these have included live-attenuated vaccines (with selected deletions), adjuvanted and unadjuvanted subunit vaccines, replication-defective vaccines (again with selected deletions and/or mutations), and adjuvanted or unadjuvanted DNA vaccines [65]. Candidate vaccines for HSV have generally failed fairly early on in phase 2 trial or earlier. Only gD2 subunit vaccines have completed large phase 3 trials (by different companies and using different adjuvant systems), but provided limited protection at best. No HSV vaccine has been licensed by the FDA. 

One problematic issue with evaluating prophylactic vaccines against HSV is that (unlike with animal models) an unambiguous history of HSV infection in humans can only be established using type-specific serology. At least half of HSV infections are asymptomatic. A US seroprevalence study published in 2014 found that 65% of people were HSV-1 seropositive by the age of 49, and 26% were HSV-2 seropositive [66]. In a study of 29,000 US women who were unaware if they had an HSV infection, 57% were infected with one or both viruses [67]. While a phase 3 trial of 30,000 participants is not particularly daunting for the performance of modern semi-automated serologic methods, the very high prevalence of both symptomatic and asymptomatic HSV prevalence complicates the screening and recruitment of a sufficient number of uninfected participants for a phase 3 clinical trial. It will be necessary to exclude adults with a history of HSV infection or, alternatively, to include four different subgroups in a trial (HSV-1+, HSV-2- HSV-1-, HSV-2+; HSV-1+, HSV-2+; HSV-1-, HSV-2-). Such an expanded study design would likely require a prohibitively large number of participants, making it effectively practical to recruit only persons seronegative for both viruses. Pre-existing immunity to either or both viruses might be expected to blunt the efficacy of vaccine-induced immunity. 

Another issue is that it is probably unrealistic to expect a vaccine to impose sterile immunity against HSV infection, since infection with HSV-2 occurs commonly in persons latently infected with HSV-1 and reinfection with both viruses can occur [68]. There is extensive evidence that all herpesviruses have a history of intra-strain recombination; thus, not only reinfection occurs but superinfection of the same cell occurs with some frequency [69]. Existing neutralizing antibodies are expected to blunt the development of lesions, but almost certainly do not prevent entrance into peripheral nerves and the establishment of neuronal latency. It has also been understood for decades that HSV can symptomatically reactivate despite high levels of circulating antibody [70,71]. This brings the conversation to the most relevant consequence of being HSV infected, namely recurrent lesions. Recrudescent HSV disease normally results from the reactivation of latent infection rather than from reinfection. Accordingly, any control of HSV using vaccines might be applied with most benefit by devising vaccines that reduce the frequency and tissue damaging effects of reactivation from latency. We argued previously that the mRNA vaccine approach might be useful in this regard by redirecting the quality of the immune response and incorporating additional components that can recruit and activate immune components responsible for lesion resolution.

### 2.4. What Differences Account for the Difficulty in Developing Effective HSV Vaccines Contrasted with the Success of VZV Vaccines?

This raises the issue of why prophylactic vaccines have achieved striking success against the cousin virus VZV in stark contrast to HSV vaccine efforts. A logical place to begin is by delineating some of the salient differences between the two viruses (Table 1).

VZV is a highly prevalent virus in temperate climates such as the US; history of VZV infection is higher than HSV-1 and HSV-2 combined [72]. VZV seroprevalence data for the US prior to the implementation of the varicella vaccine revealed that 96% of adults 20–29 years of age, 99% of persons >30 years old had serologic evidence of VZV infection [73]. Varicella vaccine coverage is more than 90% and outbreaks of varicella are now practically unknown, although this situation could change if vaccine hesitancy catalyzed by the COVID-19 pandemic, becomes more common. A study conducting active VZV statewide outbreak surveillance in six states managed to identify only a single substantial outbreak in one state over the 2-year duration of the study [74], (unpublished data). However, since the first dose of varicella is given at between 12 and 18 months of age, most recipients remain VZV seronegative at the time of inoculation. Primary infection with VZV leads to a transient T cell-associated viremia that in turn results in a generalized skin rash unlike HSV, which is usually local [75]. The transient viremic phase of VZV infection might be more relevant to T cell and antibody remediation if vaccination adequately induces these responses and establishes long-term memory. 

VZV invariably sets up latency and this usually involves more peripheral nerve ganglia than is usually the case with HSV. Given the apparently disparate rates of reactivation from latency for VZV and HSV, it may be that peripheral ganglia provide a more favorable site for maintaining latency, with limited leakage requiring ganglionic immunological activity. It is also well known that reactivation in the case of VZV usually involves relatively few latently infected ganglia. Possibly when reactivation does occur the slower replication kinetics of VZV compared to HSV provides a longer window of opportunity for immune defenses to be marshalled and limit the extent of viral replication. This concept is reinforced by the fact that providing vaccines to latently infected individuals dramatically diminishes their likelihood to suffer the recurrent lesions of shingles (a risk reduction of 97% for the recombinant adjuvanted subunit vaccine (RSV)) [76]. Finally, it is also possible that the virus itself has evolved a more robust system for the establishment and maintenance of latency, making it more refractory to reversal.

### 2.5. What Factors Have Contributed to the Success of VZV Vaccines?

The availability of two licensed and approved vaccines for the prevention of shingles afforded a unique opportunity to conduct extensive evaluations of the cell-mediated and humoral immune responses to each vaccine. This was particularly useful in this instance because the efficacy of the two vaccines was very different. A cohort study of several hundred participants examined cytokine responses in multiple subsets of T cells, both CD4+ and CD8+, as well as other measures of the cell-mediated response over a five-year period post-immunization [77,78,79]. The humoral immune responses were evaluated beyond simply measuring virus-specific IgG levels, including IgG avidity to both purified gE and to a mixture of all VZV glycoproteins, and neutralizing antibody titers [80]. Effector and memory T cell responses to recombinant vaccine were more than 10-fold higher than in those receiving the live-attenuated vaccine. The avidity results revealed that nearly all of the participants in the study had moderate to low avidity to gE, either due to a waning of high-affinity antibodies or a failure to attain high levels when primary infection occurred. Nearly every recipient of the recombinant vaccine experienced boosts in gE IgG avidity to the maximum level measurable by the assay, compared with largely unremarkable boosts from live vaccine recipients. For neutralizing antibody, the average increases in titer for live virus recipients was eight-fold, compared with 22-fold for recombinant vaccine recipients. Since affinity maturation is crucially dependent on specific forms of T cell help in the germinal centers, these results also reflect a preferential boosting of B cell memory by recombinant vaccine [81,82]. This was also reinforced by the duration of observed boosting. At the end of five years, most of the boosts in avidity due to live virus vaccination had returned nearly to baseline, whereas most recombinant vaccine recipients retained 70–80% of their boosts [80]. These aspects of the responses to the two vaccines would have gone unnoticed while the vaccines were in clinical trial. Broader studies of the immunologic responses to vaccines would be expected to provide more robust guidance on decisions to proceed with large-scale advanced vaccine trials. 

Many herpes virologists were surprised, even skeptical when the press release describing the outcome of the recombinant shingles vaccine phase 3 trial appeared. The CDC was visited in 1987 by Jonas Salk, who asked about the prospects for a subunit vaccine. At that time, he was interested in HIV, another problem child from a vaccine standpoint that might benefit from this discussion. The answer he got was that there was little percentage in anticipating that less would be more. Now along comes the recombinant adjuvanted subunit vaccine. How on Earth was it possible that this vaccine that included only a single (albeit highly immunogenic) protein from VZV so greatly outperformed a live-attenuated vaccine? Attenuated vaccines were an effective solution for the prevention of measles, mumps, rubella, varicella, polio, smallpox, rotavirus and yellow fever. What accounts for the difference in these two shingles vaccines?

One obvious contributing factor is the adjuvant system, ASO1. ASO1 is a liposome-based adjuvant with two immunostimulatory components: MPL is a detoxified derivative of lipopolysaccharide A of *Salmonella* Minnesota and QS-21 is a saponin obtained from the South American soap bark tree *Quillaja Saponaria*. Upon injection, MPL induces a broad innate immune response, transiently activating a variety of antigen-presenting cells. MPL and QS-21 operate synergistically to both stimulate and modulate both T and B cell responses [83]. It is believed to be involved in skewing the immune response toward long-term effector and memory. The downside is that the adjuvant system is quite reactogenic. The effects are transient and largely confined to injection site swelling, but can also cause fatigue, muscle pain, headache, shivering, fever and nausea.

That said, it seems unlikely that the ASO1 adjuvant alone accounts for the “less is more” aspect of the recombinant vaccine. Why would a single viral antigen induce a more effective immune response than live-attenuated vaccine, since the latter enters and replicates in cells, and probably engages a more complete T cell response including both CD4+ and cytotoxic CD8+ responses? The answer may reside in the capacity of most viruses to defend themselves by evading the immune response. Herpesviruses are the uncontested monarchs of immune evasion. Varicella has at least six immune evasion mechanisms at its disposal; cytomegalovirus at last count encodes at least 30 proteins involved in impairing multiple facets of the immune response. Vaccines that offer only a single viral protein (or several proteins) as an immunogen deprive the virus of its arsenal, freeing the immune response to respond without impairment. This possible trick for bypassing viral immune evasion tactics may be particularly important for controlling herpesvirus infections: (1) the zoster vaccines are limited to inducing components of the immune system; (2) since herpes zoster is a symptomatic reactivation of a pre-existing VZV infection from latency, an effective immune response must be capable of stabilizing the latent state and/or containing a reactivation event, and finally (3) the live-attenuated zoster vaccine—with intact immune evasion mechanisms—is far less effective at preventing zoster than the adjuvanted subunit vaccine. Nucleic acid vaccines previously had a disappointing history, proving immunogenic in small animal models but failing to scale up to humans. With the advent of the COVID-19 mRNA vaccines in 2020, all of that changed. Two highly effective vaccines were developed independently and simultaneously by different pharmaceutical firms. It should be readily apparent that such vaccines potentially offer great advantages. A carefully designed mRNA could offer both the advantage of fully engaging the T cell response (protein translation occurs necessarily in cells) and would be as free of viral anti-immune weaponry as a subunit protein vaccine.

## 3. Coda

Vaccines have had a remarkable impact on public health. Smallpox has been eradicated from the globe, and polio eradication is within reach. Measles and rubella have been eliminated from the Western hemisphere and in developed countries in other parts of the world. Varicella incidence has been reduced by over 90% in the US and varicella outbreaks are as uncommon as hen’s teeth. Illnesses that were considered a rite of childhood passage 60 years ago are now practically unheard of.

Unfortunately, however, organized opposition to vaccination has been a countervailing force for centuries, beginning with the practice of variolation in the 16th century—a practice originating in China that involved inoculating people with dried, aged and powdered material obtained from mild cases of smallpox. Vaccines can produce adverse events and these can provide the necessary fodder to fuel vaccine resistance, typically accompanied by advocacy for scientifically unproven (to say nothing of ridiculous) alternative treatments. The production of insufficiently safety-controlled batches of polio vaccine by Cutter Laboratories very nearly torpedoed the polio vaccination program. Thankfully, the program continued and polio incidence was reduced in the US by 90% in only 5 years. A falsified report by an unscrupulous and self-serving British physician falsely claimed a link between the measles vaccine and autism. Even after the physician was stripped of his license to practice medicine and several large studies unequivocally demonstrated that no such association exists, the claim that measles vaccine causes autism still persists today. This sort of resistance can have a devastating impact on public confidence in vaccines, potentially increasing the number of susceptibles in the population and causing diseases that have been brought under control to reemerge. Most recently, vaccine hesitancy regarding the COVID-19 vaccines has unnecessarily delayed and complicated the emergence from this pandemic.

The development of a highly effective vaccine, such as the recombinant adjuvanted shingles vaccine, should not automatically be accepted as perfect. The reactogenicity of that vaccine has resulted in some hesitancy in receiving the shot, despite the logical argument that a few days of discomfort are far preferable to developing shingles. The live-attenuated varicella vaccine had dramatically reduced the incidence of varicella, but the vaccine virus can establish latency. Among children, shingles is rare but does occur. One study documented cases of childhood shingles caused by the vaccine strain, although those cases occurred about one-quarter as frequently as shingles due to the wild type virus [84]. It is still too early to know how frequently vaccine shingles will occur in persons over the age of 50.

The point is an argument can be made that addressing the shortcomings of successful approved vaccines may have a positive impact on the public perception and acceptance of vaccines. For example, it might be possible to devise a shingles subunit vaccine that utilizes a less reactogenic adjuvant but provides equivalent protection to the currently approved vaccine. The expense of ushering a vaccine through the evaluation and approval process is a non-trivial obstacle to a philosophy of continuous vaccine improvement. The current road map to vaccine development and approval has placed pharmaceutical firms in the position of trying to recoup their investment by pricing vaccines at levels that are likely to discourage uptake. Multiple considerations must be taken into account in future vaccine development strategies, perhaps even obtaining competitive grant-based support for developmental phases.

So, the future of vaccine development looks bright, but special consideration must be given to the tough nuts such as HIV and HSV. Firstly, reliance on the conventional limited protocols for evaluating the immune response to prospective vaccines will likely need to be abandoned for problematic pathogens, opting instead for a much more comprehensive and granular evaluation of innate, cell-mediated and humoral immunity. It may well also be that we will need to lower our expectations with respect to viruses that are able to reinfect people and that do a particularly good job of hiding themselves from harm. Outright prevention of HSV infection is almost certainly an unobtainable goal. But consider that the most relevant consequence of being HSV infected is recurrent lesions or silent viral shedding. Latently infected persons can experience either manifestation; conceivably, boosting or reshaping the nature of adaptive immunity could be effectively directed to the goal of reducing the rate and duration of shedding episodes or possibly prevent shedding altogether. Such immunological management would be greatly appreciated by those many persons who experience frequent problems with recurrent herpetic lesions, especially when these affect the eye and genitalia. Rather than relying on a single viral mRNA, there should be no obstacle to designing vaccines with multiple mRNAs, broadening the immune response without compromising the advantage of disarming viral defense mechanisms. New experimental approaches to determine whether vaccines early in development are addressing the targeted outcome measures would also be important.

In conclusion, new technologies offer hope that vaccines effective at reducing disease for agents that have proven refractory to prevention appear to be on the horizon. Regrettably, the authors of this review will not be engaged in the process of developing them, as we have retired from the game!

## Figures and Tables

**Table 1 viruses-14-00862-t001:** Comparison of characteristics of VZV and HSV.

VZV	HSV	Caveats
Primary infections are generally asymptomatic	About half of primary infections are asymptomatic	None
Transmission by aerosol droplets or contact with vesicles	Transmission through physical contact with active shedder	Some evidence of remote transmission through air handling systems
Generalized pruritic vesicular rash (centripetal distribution)	Typically a single local lesion; painful but not pruritic	Primary HSV infection can be occasionally viremic
Reactivates infrequently	Reactivates often	Some VZV reactivations may be missed
Extremely fastidious. Infects primarily T lymphocytes, neurons, epithelial cells; less efficiently Vero cells, guinea pig embryonic cells	Infects a broad variety of cells and animals	None
No animal models; SVV model in African green monkeys and macaques	Animals are available but do not generally replicate human disease	HSV-2 guinea pig genital herpes model approximates human disease

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
