# Peer review of "Fraternal Twins: The Enigmatic Role of the Immune System in Alphaherpesvirus Pathogenesis and Latency and Its Impacts on Vaccine Efficacy"

_viruses, 2022, doi:10.3390/v14050862_

Round 1
Reviewer 1 Report
The paper by Rouse et al., reviews the current understanding of HSV, and VZV latency; discusses the characteristics of HSV in comparison to that of VZV that potentially explains lack of effective vaccines against HSV. The authors also share their opinion on how effective HSV vaccines can be developed.
Overall, I think the review article is very well written clarifying the current state of knowledge, identifying the gaps, and directing the areas where research is much needed. I have some minor suggestions as below-
- Please define the acronym for VZV in the abstract and both HSV, and VZV in the introduction.
- Line 49- "viral mutants unable to produce the LAT transcripts can still establish and maintain latency in animal models". Please provide any known information on what type of viral genes are expressed in these mutants that could potentially be contributing towards establishment and maintenance of latency.
- Line 90- as the role of innate immunity in HSV pathogenesis is first introduced here, I suggest mentioning the type of innate immune cells such as macrophages, or perhaps Langerhan cells that are known to be most critical for dealing with HSV infection.
- Line 191- please mention a few examples of the stimuli that cause HSV reactivation.
- Line 246- have researchers looked into the chemokine expression profiles of VZV-infected ganglia which leads to infiltration of macrophages and NK cells? If yes, you could add their findings here with references.
- Line 331- please add a sentence on what factors including immune cell types make up the microenvironment of ganglia.
Author Response
(x) I would not like to sign my review report
( ) I would like to sign my review report
English language and style
( ) Extensive editing of English language and style required
( ) Moderate English changes required
(x) English language and style are fine/minor spell check required
( ) I don't feel qualified to judge about the English language and style
|
Yes |
Can be improved |
Must be improved |
Not applicable |
|
|
Does the introduction provide sufficient background and include all relevant references? |
(x) |
( ) |
( ) |
( ) |
|
Is the research design appropriate? |
( ) |
( ) |
( ) |
(x) |
|
Are the methods adequately described? |
( ) |
( ) |
( ) |
(x) |
|
Are the results clearly presented? |
(x) |
( ) |
( ) |
( ) |
|
Are the conclusions supported by the results? |
( ) |
( ) |
( ) |
(x) |
Comments and Suggestions for Authors
The paper by Rouse et al., reviews the current understanding of HSV, and VZV latency; discusses the characteristics of HSV in comparison to that of VZV that potentially explains lack of effective vaccines against HSV. The authors also share their opinion on how effective HSV vaccines can be developed.
Overall, I think the review article is very well written clarifying the current state of knowledge, identifying the gaps, and directing the areas where research is much needed. I have some minor suggestions as below-
- Please define the acronym for VZV in the abstract and both HSV, and VZV in the introduction. DONE
- Line 49- "viral mutants unable to produce the LAT transcripts can still establish and maintain latency in animal models". Please provide any known information on what type of viral genes are expressed in these mutants that could potentially be contributing towards establishment and maintenance of latency.
Published data on LAT- minus (or partial deletions of the transcript or promoter) are overall inconsistent. Many report that the number of latently infected cells is reduced 3-5-fold over strains with intact LAT; others show no difference in the rate of establishing latency. More consistent (but not completely so) are observations that LAT mutants have reduced or completely ablated capacity to reactivate. Another relatively consistent observation is that cells latently infected with LAT mutants are leaky with respect to protein expression. While this is an interesting discussion, our feeling is that it diverts us too much from the stated goals of the review and would require too much detail to do justice to the subject. Our preference is not to go there.
- Line 90- as the role of innate immunity in HSV pathogenesis is first introduced here, I suggest mentioning the type of innate immune cells such as macrophages, or perhaps Langerhan cells that are known to be most critical for dealing with HSV infection. Amended as follows: These innate responses usually succeed in controlling the infection and resolving lesions, including interferons, neutrophils, macrophages, dendritic cells and natural killer cells [10]. Added new references and renumbered references.
- Line 191- please mention a few examples of the stimuli that cause HSV reactivation. Added the following text: Potential triggers for reactivation include emotional stress, hormonal changes, UV light exposure, among others.
- Line 246- have researchers looked into the chemokine expression profiles of VZV-infected ganglia which leads to infiltration of macrophages and NK cells? If yes, you could add their findings here with references. This is discussed in some detail in the paragraph starting on line 286 of the revised manuscript.
- Line 331- please add a sentence on what factors including immune cell types make up the microenvironment of ganglia. This is also discussed in the paragraph beginning on line 286.
Submission Date
06 April 2022
Date of this review
15 Apr 2022 04:15:10

Reviewer 2 Report
This review is timely and well constructed. The style is at times unusual for a scientific publication, but these instances mostly do not detract from meaning. Some phrasing may be overstated and these instances are indicated below. There are some typographical errors, many of which are pointed out below:
is Line 31. “Guru” is an unconventional term with connotations that may go beyond the intent here. “Longstanding expert” may be more appropriate.
- Although this reviewer appreciates the Steinbeck reference, the use of “men” here is overly restrictive, as the rule also applies to women. Elsewhere, the word “mankind” could be replaced by “humankind” “people” or “humans”.
59, 60. It is not yet clear whether there is a complete absence of viral proteins in VZV latency because the experiment to determine validity cannot be done readily. The cited reference 6 does not prove the absence of proteins in VZV infected ganglia. Presumably this is supposed to be reference 41. Removing the “artifact” statement and stating the obvious caveat of the cadaver study seems more appropriate here and better represents the consensus.
- The immune system…
- HHV3 should appear at first mention of VZV. HHV1 should appear at the first mention of HSV-1 etc.
- … as well as…
- Interrupting axonal transport might very well be more dangerous than any virus for which therapy is proposed.
Line 118. Some HSV-infected neurons may never reactivate in a lifetime, so is that not permanent latency?
Line 149. Period in wrong place.
- Not sure why this has to be a fraternity. Why not a group or sorority?
211-217. A bit unclear. This is not a vaccine but gene therapy. Are viral antigens to be included in these novel mRNA pools?
- Are the mice infected or is the implant infected?
Line 269 switch “also” and “the”.
Line 264. Were all proteins ruled out because of blood group antigen cross reactivity, or just a subset? The stated reference (41) looked at only 3 antisera.
Line 286. Insert “from”.
Line 376. Remove one “prevalence”.
Line 492-493. I do not understand this sentence.
Line 514. “Undisputed monarchs” may be overstated. Poxviruses would dispute the claim.
Author Response
(x) I would not like to sign my review report
( ) I would like to sign my review report
English language and style
( ) Extensive editing of English language and style required
(x) Moderate English changes required
( ) English language and style are fine/minor spell check required
( ) I don't feel qualified to judge about the English language and style
|
Yes |
Can be improved |
Must be improved |
Not applicable |
|
|
Does the introduction provide sufficient background and include all relevant references? |
( ) |
( ) |
(x) |
( ) |
|
Is the research design appropriate? |
( ) |
( ) |
( ) |
(x) |
|
Are the methods adequately described? |
( ) |
( ) |
( ) |
(x) |
|
Are the results clearly presented? |
( ) |
( ) |
( ) |
(x) |
|
Are the conclusions supported by the results? |
(x) |
( ) |
( ) |
( ) |
Comments and Suggestions for Authors
This review is timely and well constructed. The style is at times unusual for a scientific publication, but these instances mostly do not detract from meaning. Some phrasing may be overstated and these instances are indicated below. There are some typographical errors, many of which are pointed out below:
is Line 31. “Guru” is an unconventional term with connotations that may go beyond the intent here. “Longstanding expert” may be more appropriate. Changed to “leading HSV authority”
- Although this reviewer appreciates the Steinbeck reference, the use of “men” here is overly restrictive, as the rule also applies to women. Elsewhere, the word “mankind” could be replaced by “humankind” “people” or “humans”. Changed to “humans”
59, 60. It is not yet clear whether there is a complete absence of viral proteins in VZV latency because the experiment to determine validity cannot be done readily. The cited reference 6 does not prove the absence of proteins in VZV infected ganglia. Presumably this is supposed to be reference 41. Removing the “artifact” statement and stating the obvious caveat of the cadaver study seems more appropriate here and better represents the consensus. Agreed. Since this is discussed in more detail later in the manuscript, we replaced the statements with the following text: The discovery of varicella latency transcript (VLT) implied that VZV latency probably more closely resembles that of HSV [6, 7]. Whether or not any viral proteins are produced during VZV latency remains to be formally demonstrated.”
We also swapped the positions of reference 6 and 7.
- The immune system…DONE
- HHV3 should appear at first mention of VZV. HHV1 should appear at the first mention of HSV-1 etc. These designations are almost never used for HSV-1, HSV-2, VZV, EBV and CMV. It was a well-intended effort but has just not ever caught on. In our opinion introducing those terms here just doesn’t add anything of value. Not sure what the impulse was to use the alternative acronym HHV3 for VZV was, but we removed it.
- … as well as…DONE
- Interrupting axonal transport might very well be more dangerous than any virus for which therapy is proposed. We agree, but we haven’t really proposed using a therapeutic strategy that interrupts axonal transport. The argument that vaccines may either prevent reactivation from latency or to contain reactivation was not meant to imply that they interrupt axonal transport. That seems highly improbable. More likely such containment would at the site of exonal exit into the skin, preventing the formation of lesions.
Line 118. Some HSV-infected neurons may never reactivate in a lifetime, so is that not permanent latency? The physiological triggers for reactivation never lead to the reactivation of every last latently infected neuron. The point here is that partial reactivations appear to occur leading to some transient viral protein expression before locking down latency again. The possibility that some latently infected neurons never reactivate in no way implies that they are incapable of undergoing reactivation from latency. This strikes us as kind of a specious argument.
Line 149. Period in wrong place. CORRECTED
- Not sure why this has to be a fraternity. Why not a group or sorority? Changed to “constituency”
211-217. A bit unclear. This is not a vaccine but gene therapy. Are viral antigens to be included in these novel mRNA pools? Yes, of course. But we would argue against the declaration that these vaccines represent gene therapy. Gene therapy is generally aimed at making changes to the somatic host genome (albeit such approaches are currently pretty limited in scope). mRNA vaccines involve the transformation of viral mRNAs that encode specific key immunogenic viral proteins. Like all mRNAs they have limited half-lives and survive in cells only long enough to produce sufficient protein to generate a robust immune response. A chief advantage of this strategy is that, since the protein is produced within cells, both CD4 and CD8 T cell responses are engaged. The point of multiple mRNAs is to produce more than one viral protein target for the immune response.
- Are the mice infected or is the implant infected? Kind of a matter of semantics. The implant (located in the mouse) is infected. Changed from mice to “implanted DRG”
Line 269 switch “also” and “the”. DONE
Line 264. Were all proteins ruled out because of blood group antigen cross reactivity, or just a subset? The stated reference (41) looked at only 3 antisera. You are correct We have changed the text as follows:
“To a lesser extent, the ORF63 gene is also transcribed in latently infected neurons, but no other VZV transcripts and no VZV proteins are detectable. Early reports of multiple VZV proteins in latently infected cells were called into question when it was discovered that several ascites-derived anti-VZV reagents cross-react with the blood group A antigens [42]. Moreover, analysis of the complete VZV transcriptome in trigeminal ganglia obtained immediately post-mortem among VZV latently infected subjects produced no viral transcripts for up to 9 hours post-mortem (39). Thus, current observations imply that expression of LAT and shutdown of viral transcription are consistent among the human alphaherpesviruses.
Line 286. Insert “from”. DONE
Line 376. Remove one “prevalence”. DONE
Line 492-493. I do not understand this sentence. Changed as follows: Broader studies of the immunologic responses to vaccines would be expected to provide more robust guidance on decisions to proceed with large-scale advanced vaccine trials.
Line 514. “Undisputed monarchs” may be overstated. Poxviruses would dispute the claim. We respectfully disagree. No other family of viruses (including HPV and HIV) have developed an immune evasion tactic as efftective as the life-long relationship established using latency. Every human herpesvirus has evolved multiple mechanisms for impairing the immune response in addition to latency. Thus far, at least 30 CMV-encoded proteins are involved in evading host immunity, including proteins that individually impair every step of antigen processing. Poxviruses, like most DNA viruses have indeed evolved mechanisms for impairing host immunity, but poxvirus infections (if they don’t kill you) are effectively cleared by the immune system. Sterilizing immunity is the rule for poxviruses. Sterilizing immunity is, at least thus far, an impossible outcome for herpesvirus infections. They are without exaggeration, in our opinion, the undisputed masters of immune evasion.
Submission Date
06 April 2022
Date of this review
06 Apr 2022 19:46:22
